# Estimation of neuron parameters from imperfect observations

**Joseph D. Taylor**[ID], **Samuel Winnall**[ID], **Alain Nogaret**[ID] *

Department of Physics, University of Bath, Bath, United Kingdom

* A.R.Nogaret@bath.ac.uk

## Abstract

The estimation of parameters controlling the electrical properties of biological neurons is essential to determine their complement of ion channels and to understand the function of biological circuits. By synchronizing conductance models to time series observations of the membrane voltage, one may construct models capable of predicting neuronal dynamics. However, identifying the actual set of parameters of biological ion channels remains a formidable theoretical challenge. Here, we present a regularization method that improves convergence towards this optimal solution when data are noisy and the model is unknown. Our method relies on the existence of an offset in parameter space arising from the interplay between model nonlinearity and experimental error. By tuning this offset, we induce saddle-node bifurcations from sub-optimal to optimal solutions. This regularization method increases the probability of finding the optimal set of parameters from 67% to 94.3%. We also reduce parameter correlations by implementing adaptive sampling and stimulation protocols compatible with parameter identifiability requirements. Our results show that the optimal model parameters may be inferred from imperfect observations provided the conditions of observability and identifiability are fulfilled.

**Data Availability Statement:** All relevant data are within the manuscript and its Supporting Information files.

**Funding:** This work was supported by the European Union's Horizon 2020 Future Emerging Technologies Programme under Grant No.

## Author summary

The accurate estimation of neuronal parameters inaccessible to experiment is essential to our understanding of intracellular dynamics and to predicting the behaviour of biocircuits. However, this program is met with challenges including our lack of knowledge of the precise equations of biological neurons, their highly nonlinear response to stimulation and error introduced by the measurement apparatus. The imprecise knowledge of model and data introduces uncertainty in the parameter field. Our work describes a regularization method that arrives at the optimal parameter solution with a probability of 94%. The uncertainty on parameter estimates is further reduced with the help of an adaptive sampling method that maximises the duration of the assimilation window while keeping the size of the problem constant. Our work shows that the true configuration of a neuronal system may be inferred from time series observations provided external stimuli are calibrated to drive the system over its entire dynamic range.

732170. JDT acknowledges the EPSRC(UK) for a DTP studentship award. SW was partially supported by the Institute of Mathematical Innovation at the University of Bath. The funders had no role in study design, data collections and analysis, decision to publish, or preparation of the manuscript.

This is a *PLOS Computational Biology* Methods paper.

## Introduction

Data assimilation is increasingly important in quantitative biology to infer unmeasurable microscopic quantities from the observation of macroscopic variables. It has successfully obtained quantitative neuron models by synchronizing model equations to membrane voltage oscillations [1–3] and inferred the connectivity of neuron populations from electroencephalographic recordings of brain activity [4, 5]. Models constructed from time series analysis have been reported to accept multi-valued parameter solutions [6, 7]. The identification of the optimal solution, among all others producing equivalent outcomes, is currently a road block on the way to resolving the phenotype of neurons and biocircuits. A different, yet related problem, is that, under ordinary conditions, biocircuits may exhibit functional overlap [8, 9], redundancies [10] and compensation [11]. This further increases the need to determine whether experimental protocols exist which can yield actual biocircuit parameters. Criteria for identifying the true parameters of such systems would allow classifying neuronal phenotypes [12, 13], unknown cell types [2, 14], and understanding the effect of channelopathy on neuron dynamics [15] in Alzheimer's disease [16–18], seizures [19, 20], and Parkinson's disease [15, 21]. We now briefly review the theoretical challenges of estimating parameters with inverse methods before summarizing our solutions.

Neuron-based conductance models are described by nonlinear differential equations:

$$\dot{\boldsymbol{x}}(t) = \boldsymbol{F}[\boldsymbol{x}(t), \boldsymbol{p}, \boldsymbol{I}_{inj}(t)]. \tag{1}$$

The $x_1(t), \ldots, x_L(t)$ are the state variables including: membrane voltage, ionic gate variables, synaptic currents; the $p_1, \ldots, p_K$ are model parameters; and $\boldsymbol{I}_{inj}(t)$ is the control vector whose components are the current protocols injected in one or more neurons. Takens' embedding theorem states that information about a dynamic system is preserved within the time series recording of its output over a finite duration [22, 23]. This warrants the existence of a unique parameter solution provided the following conditions are satisfied:

- *Observability*

  The system modelled by Eq 1 is observable if its *initial conditions* can be estimated from observations of its state dynamics over a finite time interval [24–26]. If the neuron membrane voltage, $V_{\exp}(t)$, is the state variable being measured, one defines a measurement function $V_{\exp}(t) = h(x_1(t), \ldots, x_L(t), p_1, \ldots, p_K) = x_1(t)$ which relates $V_{\exp}(t)$ to the $L$-dimensional state vector $\mathbf{x}$ and the $K$-dimensional parameter vector $\boldsymbol{p}$. Since parameters may be viewed as constant state variables satisfying $\dot{\boldsymbol{p}} = 0$, the state of the system is a $L + K$-dimensional vector. A single measurement of $V_{\exp}$ at time $t$ however does not contain all the information needed to determine all vector components. The missing information may be recovered by constructing an $L + K$-dimensional embedding vector that is either based on the derivatives of the observed state variable $x_1(t), \ldots, x_1(t)^{(L+K)}$ or its delay coordinates $x_1(t), \ldots, x_1(t - (L + K)\tau)$. This vector is then embedded in the time series $V_{\exp}(t), \ldots, V_{\exp}^{(L+K)}(t)$ or $V_{\exp}(t), \ldots, V_{\exp}(t - (L + K)\tau)$ respectively. Takens' theorem specifies that the embedding space must have at least $2(L + K) + 1$ samples for the system to be observable [22, 23, 27] although simulations by Parlitz et al. [25, 26] have shown that an embedding space equal to the number of state variables is generally sufficient. The time series which are assimilated usually hold $n = 10, 000 - 100, 000$ data points [1–3] which amply fulfill the observability requirement, $n \gg 2(L + K) + 1$, if

$L + K < 100$ typically. Twin experiments have verified that the assimilation of large data sets [28–30] infers the original model parameters of well-posed problems [31].

- *Identifiability*
  Any two pairs of parameter sets $\boldsymbol{p}_1 \neq \boldsymbol{p}_2$ are identifiable if they result in different state trajectories $\boldsymbol{x}_1(t) \neq \boldsymbol{x}_2(t)$ given the same driving force, $I_{inj}(t)$, and same initial conditions $\boldsymbol{x}_1(0) = \boldsymbol{x}_2(0)$. Parameter identifiability is highly dependent on the choice of driving force [32]. However, the driving force criteria that make parameters identifiable have not been studied so far, partly because most investigations have focused on self-sustaining oscillators [8, 33].

- *Local minima in the cost function*
  Variational cost functions are often riddled with local minima [34] giving sub-optimal parameters solutions. The probability of parameter search arriving at such false solution is enhanced by the presence of experimental error particularly when this error becomes comparable or greater than the error introduced by sub-optimal parameters. In this situation, minimizing the cost-function alone is unable to resolve optimal from sub-optimal parameter solutions. A regularization method is thus needed to recover the optimal solution.

- *Ill-defined problems*
  The model equations of biological neurons are unknown [1, 2]. The guessed conductance models carry model error whose effect on parameter solutions needs evaluating. Secondly unknown models carry the risk of over-specifying ion channels and failing to meet identifiability criteria [5, 35, 36].

Here we address the problem of multi-valued solutions in the optimization of neuron-based conductance models. The effects of experimental and model error on these solutions is demonstrated from general considerations on the cost function. We then use an exemplar conductance model to demonstrate the enhancement of convergence towards the optimum parameter solution. The model is a variant of the multichannel conductance models which were proven to successfully assimilate biological neurons ranging from songbird neurons [1, 2] and hippocampal neurons [3, 37] to respiratory neurons [3]. The exemplar model displays the same multiplicity of sub-optimal solutions encountered in all neuron-based conductance models including those derived from Hodgkin-Huxley equations [1, 2, 37, 38] or analog device equations [3, 39]. We began by performing random Monte-Carlo simulations of the posterior distribution function (PDF) of model parameters estimated from noisy data. We show that the interplay of model nonlinearity, experimental error and model error, shifts the maximum likelihood expectation (MLE) and standard deviation of estimated parameters. The realization of noise across the measurement window is found to shift the location of the local and global minima relative to one another on the data misfit error surface. Experimental error also tilts the principal axes of surfaces of constant data misfit error centered on each minimum. We use these findings to regularize convergence towards the optimum parameter solution when parameter search would otherwise stop at a local minimum near the global minimum. This novel method increases the probability of convergence towards the true global minimum from 67% to 94%. We also reduced the correlations between parameters by over an order of magnitude by increasing the duration of the assimilation window while keeping the size of the problem constant. For this we introduced an adaptive sampling rate which applied a longer time step during intervals of sub-threshold oscillations. Our simulations also show that models configured with sub-optimal parameters output membrane voltage oscillations which are always distinguishable from those of models configured with optimal parameters. Hence even biocircuits exhibiting functional overlap under normal conditions [6, 8, 9, 40] may have their parameters fully

determined under appropriate external stimulation with the regularization method we introduce here.

The paper is structured as follows. The *first section* describes the effects of experimental error and model error on the data misfit surface. We calculate the parameter offset $\delta\boldsymbol{p}_{\sigma\zeta}$ as a function of the amplitude ($\sigma$) and realization ($\zeta$) of additive noise and model error. The *second section* computes the posterior distribution functions of extracted parameters and investigates their shape, MLE and, covariance. The *third section* describes the regularization method that uses the above parameter offset to enhance the probability of convergence to the optimal parameter solution. The *fourth section* describes the adaptive sampling method we use to enhance parameter identifiability. The *last section* discusses predictions made by models configured with optimal and sub-optimal parameters. The results show that under appropriate conditions of stimulation, the oscillations produced by disparate sets of parameters are always distinguishable.

## Results

### Noise-induced shift in parameter solutions

One defines a least-squares cost function to measure the distance between the state variable of the membrane voltage in the model $V_{mod}(t_i, \boldsymbol{x}(0), \boldsymbol{p})$ and the experimentally observed membrane voltage $V_{\exp}(t_i)$. $\boldsymbol{x}(0)$ are the initial conditions of the state variables for the model. The cost function is evaluated at each mesh point $i = 0\dots n$ of the assimilation window:

$$c(\boldsymbol{x}(0), \boldsymbol{p}) = \frac{1}{2}\sum_{i=0}^{n}(V_{\exp}(t_i) - V_{mod}(t_i, \boldsymbol{x}(0), \boldsymbol{p}))^2 + u^2(t_i), \qquad (2)$$

where the $x_l(t)$, $l = 1\dots L$ are the state variables of the neuron-based conductance model and the $p_k$, $k = 1\dots K$ are the parameters of the model. State variables are evaluated at discrete times $t_i = iT/n$, $i = 0\dots n$ across the assimilation window of duration $T$. They typically include the membrane voltages, gate variables and synaptic currents of conductance models. The function $u(t)$ is a Tikhonov regularization term [41] which smoothes convergence over successive iterations by eliminating positive values of the conditional Lyapunov exponents [42]. $u(t)$ is also evaluated at discrete times like other state variables but under the constraint that it varies smoothly rather than according to Eq 1 (see Methods section).

In order to separate the contributions of experimental error and model error, we introduce the useful membrane voltage, $V_{use}(t_i)$, that is the voltage that would be measured by the ideal current clamp (Fig 1(a)). This approach allows us to separate experimental error, $\epsilon_{\exp}(t_i) = V_{\exp}(t_i) - V_{use}(t_i)$, from model error, $\epsilon_{mod}(t, \boldsymbol{x}(0), \boldsymbol{p}) = V_{mod}(t, \boldsymbol{x}(0), \boldsymbol{p}) - V_{use}(t)$. Experimental error, $\epsilon_{\exp}(t_i)$, covers patch clamp noise, thermal fluctuations, stochastic processes associated with the opening and closing of ion channels, the binding of signalling molecules to receptors, and long term membrane potentiation [43]. We model this below with $n + 1$ random variables $\epsilon_{\sigma\zeta}(t_i)$, $i = 0\dots n$, each of which follows a normal distribution, $\mathcal{N}(0, \sigma)$, with zero mean and standard deviation $\sigma$. Individual realizations of noise across the assimilation window are labelled $\zeta$. The cost function in Eq. refeq:eq1 is only suitable for uncorrelated noise. Temporally correlated noise, or more generally temporally correlated measurements, would be treated in the same way by substituting the least square cost function with a cost function incorporating an error conditioning covariance matrix [44] accounting for correlations between measurements through finite off-diagonal terms. Unlike experimental error, model error depends on the model parameters. The cost function may thus be expanded with respect

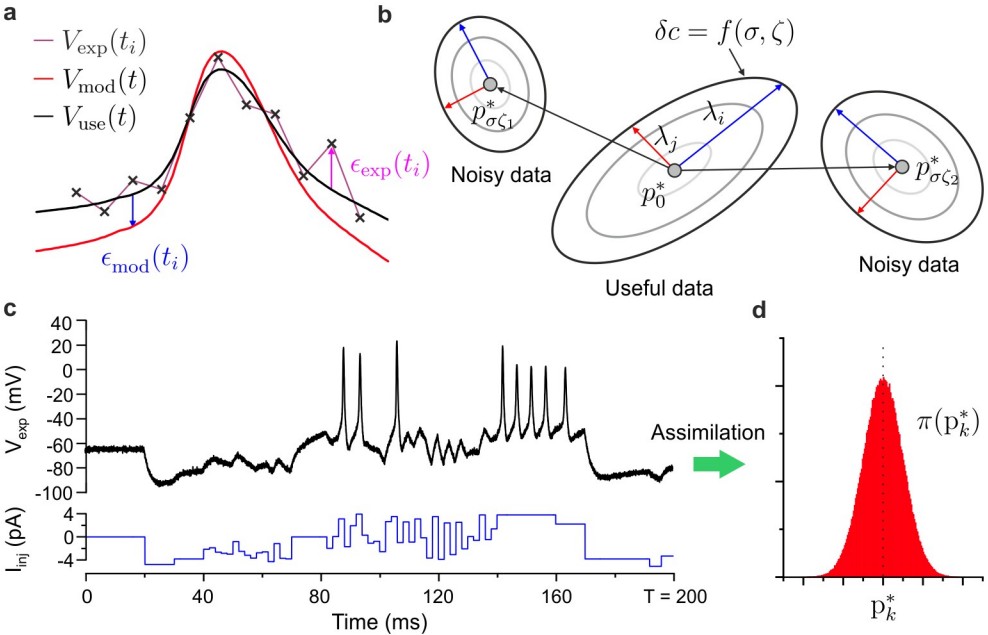

**Fig 1. Data misfit surface perturbed by experimental and model error.** (a) Membrane voltage, $V_{exp}(t_i)$, recorded in discrete time $t_i$, $i = 0 \ldots n$ (cross symbols); useful membrane voltage, $V_{use}(t)$, obtained from an ideal measurement apparatus (black line); membrane voltage state variable of the conductance model, $V_{mod}(t)$ (red line). Experimental error: $\epsilon_{exp}(t_i) = V_{exp}(t_i) - V_{use}(t_i)$. Model error: $\epsilon_{mod}(t) = V_{mod}(t) - V_{use}(t)$. (b) Lines of constant data misfit, $\delta c = f(\sigma, \zeta)$, about the global minimum $p_0^*$. Different noise realizations, $\zeta_1$ ($\zeta_2$), shift the global minimum $\boldsymbol{p}_0^* \rightarrow \boldsymbol{p}_{\sigma\zeta_1}^*$ ($\boldsymbol{p}_0^* \rightarrow \boldsymbol{p}_{\sigma\zeta_2}^*$). Noise also tilts the principal axes of the data misfit ellipsoid (red/blue arrows) and modifies the principal semi-axes ($\lambda_i$, $\lambda_j$). (c) RVLM neuron model membrane voltage $V_{exp}$ (black line) induced by current injection $I_{inj}$ (blue line). Additive noise $\epsilon_{\sigma\zeta}$ is incorporated in the model data. (d) Posterior distribution function $\pi(p_k)$ of parameter $p_k$, $k = 1 \ldots K$.

to model parameters as:

$$c(\boldsymbol{x}(0), \boldsymbol{p}) = \frac{1}{2}\sum_{i=0}^{n}\epsilon_{mod}^2(t_i, \boldsymbol{x}(0), \boldsymbol{p}) + u^2(t_i)$$
$$+ \frac{1}{2}\sum_{i=0}^{n}\epsilon_{\sigma\zeta}^2(t_i) + \sum_{i=0}^{n}\epsilon_{\sigma\zeta}(t_i)\epsilon_{mod}(t_i, \boldsymbol{x}(0), \boldsymbol{p}),$$

(3)

to separate the error contributions from model and measurements.

One now considers how perturbations of the useful signal by experimental error and model error modify the cost function in the vicinity of a local/global minimum. Labelling the true global minimum at zero noise, $\boldsymbol{p}_0^*$, we compute the data misfit $\delta c = c(\boldsymbol{x}(0), \boldsymbol{p}_{\sigma\zeta}) - c(\boldsymbol{x}(0), \boldsymbol{p}_0^*)$. This gives the perturbation of the cost function by noise. The first three terms in the expansion about the true minimum $\boldsymbol{p}_0^*$:

$$\delta c = F + (\boldsymbol{p} - \boldsymbol{p}_0^*)^T\boldsymbol{G} + \frac{1}{2}(\boldsymbol{p} - \boldsymbol{p}_0^*)^T\hat{\boldsymbol{H}}(\boldsymbol{p} - \boldsymbol{p}_0^*)\cdots$$

(4)

include the offset $F$ representing signal noise entropy, a finite gradient $\boldsymbol{G}$ arising from the interplay between model nonlinearity and the realization of noise, and the Hessian $\hat{\boldsymbol{H}}$

 

perturbed by experimental and model errors. These three terms are:

$$
\begin{aligned}
H_{kk'} &= \sum_{i=0}^{n} \frac{\partial V_{mod}}{\partial p_k} \bigg|. \frac{\partial V_{mod}}{\partial p_{k'}}\bigg|_{\boldsymbol{p}_0^*} + \frac{\partial^2 V_{mod}}{\partial p_k \partial p_{k'}}\bigg|_{\boldsymbol{p}_0^*} \left[\epsilon_{\sigma\zeta}(t_i) + \epsilon_{mod}(t_i, \boldsymbol{x}(0), \boldsymbol{p})\right], \\
G_k &= \sum_{i=0}^{n} \epsilon_{\sigma\zeta}(t_i) \frac{\partial V_{mod}}{\partial p_k}\bigg|_{\boldsymbol{p}_0^*}, \\
F &= \frac{1}{2}\sum_{i=0}^{n} \epsilon_{\sigma\zeta}^2(t_i) + \sum_{i=0}^{n} \epsilon_{\sigma\zeta}(t_i)\epsilon_{mod}(t_i, \boldsymbol{x}(0), \boldsymbol{p}).
\end{aligned}
\tag{5}
$$

The surface of constant data misfit $\delta c = f(\sigma, \zeta)$ (Fig 1(b)), is a $K$-dimensional ellipsoid. Gradient $\boldsymbol{G}$ (Eq 4) is responsible for shifting the centre of the ellipsoid from $\boldsymbol{p}_0^*$ to a new location $\boldsymbol{p}_{\sigma\zeta}^*$. This propels the new minimum to a different location in parameter space which depends on the noise realization, $\zeta$ (Fig 1(b)). The vector components of $\boldsymbol{G}$ will in general remain finite due to the interplay of model nonlinearity with noise (Eq 6). The dominant contribution to the $\partial V_{mod}/\partial p_k$ term will come from jumps in membrane voltage (-100mV $\leftrightarrow$ +45mV) near action potentials that can be induced by minute changes in parameter values. Hence, noise weighted derivatives $\partial V_{mod}/\partial p_k$ averaged across the assimilation window give finite gradient values $G_k(\zeta)$ which depend on noise realizations. Different noise realizations thus give different parameter offsets, $\delta\boldsymbol{p}_{\sigma\zeta} = \boldsymbol{p}_{\sigma\zeta}^* - \boldsymbol{p}_0^*$ (Fig 1(b)).

Before proceeding with the calculation of the parameter offset, note the superposition of noise and model error in $\hat{\boldsymbol{H}}$. The first term in $H_{k,k'}$ gives the curvature of the data misfit surface. This term determines how tightly constrained a parameter estimate is, also labelled parameter "sloppiness" by Gutenkunst et al. [7]. The second term in $H_{k,k'}$ gives the perturbation of this curvature by noise and model error. As noted above, the second derivative of $V_{mod}$ with respect to parameters $p_k$ and $p_{k'}$ weighted by error does not cancel by summation across the assimilation window. As a result, noise and model error are expect to tilt the principal axes of the ellipsoid and change their semi-axes. Experimental and model error thus alter parameter correlations.

The $F$ term represents the signal noise entropy supplemented by correlations between noise and model error. The dominant first term is the random energy $T_{\sigma\zeta}dS$ that relates to noise entropy $dS$ through the Johnson-Nyquist theorem [45, 46]:

$$
\frac{1}{2}\sum_{i=0}^{n} \epsilon_{\sigma\zeta}^2(t_i) = 2(n+1)k_B T_\sigma R \Delta f,
\tag{6}
$$

where $k_B$ is Boltzmann's constant, $R$ is the membrane resistance of the neuron, $\Delta f$ is the bandwidth of noise and $T_\sigma$ is the noise-equivalent temperature.

The noise-induced shift in $\delta\boldsymbol{p}_{\sigma\zeta}$ is obtained through principal component analysis of the Hessian matrix. In the basis of its eigenvectors, the Hessian $\hat{\boldsymbol{H}}' = \hat{\boldsymbol{V}}^T\hat{\boldsymbol{H}}\hat{\boldsymbol{V}}$ is a $K \times K$ diagonal matrix $\hat{H}' = diag(\lambda_1^{-2}, \ldots, \lambda_K^{-2})$ where the $\lambda_k$ are the principal semi-axes of the data misfit ellipsoid. $\hat{\boldsymbol{V}}$ is the $K \times K$ orthonormal matrix of eigenvectors transforming $\delta\boldsymbol{p}$ into $\delta\boldsymbol{p}' = \hat{\boldsymbol{V}}T\delta\boldsymbol{p}$ in the new basis and $\boldsymbol{G}$ into $\hat{\boldsymbol{G}}' = \hat{\boldsymbol{V}}^T\hat{\boldsymbol{G}}$. The data misfit may be written as:

$$
\delta c = F' + \sum_{k=1}^{K} \left(\delta p_k' + \frac{G_k'}{\lambda_k}\right)\frac{1}{2\lambda_k}\left(\delta p_k' + \frac{G_k'}{\lambda_k}\right),
\tag{7}
$$

where

$$F' = F - \sum_{k=1}^{K} \frac{(G'_k)^2}{2\lambda_k}. \tag{8}$$

The noise-induced offset follows from Eq 7 as $\delta p = V^T \hat{H}^{-1} G$. Through gradient $G$, experimental error gives the first order contribution to the noise-induced parameter shift (Eq 7). Model error gives a second order contribution through its perturbation of $\hat{H}$. The tilt of the principal axes of the ellipsoid is given by the eigenvectors in matrix $V$ and their semi−axes are the $\lambda_k$ eigenvalues.

## Posterior distribution function of optimal parameters

To demonstrate the above results, we now compute the effect of noise amplitude on the PDF of optimal parameters. The next section will then evaluate the parameters arising from individual noise realizations rather than a statistical ensemble and calculate individual parameter offsets relative to when no noise is applied.

We choose the conductance model of a rostral ventrolateral medulla (RVLM) neuron located at the base of the brain [47, 48]. This neuron accelerates heart rate when blood pressure increases for instance and balances the bradycardia action of vagal motoneurons [47]. The RVLM neuron has a wide complement of ion channels (Table 1), and as such is an appropriate neuron to model. The somatic compartment of RVLM neurons includes the following ion channels [48]: transient sodium channels (NaT), depolarization-activated potassium channels (K), leak channels (Leak), hyperpolarization-activated cation channels (HCN), and low threshold calcium channels (CaT). The RVLM model has 7 state variables ($L = 7$) and 41 parameters ($K = 41$). The biological parameters are the vector components of $p_{true}$ in Table 2. Model data, $V_{use}(t)$, were then synthesized by using the RVLM model configured with $p_{true}$ to forward integrate the current protocol of Fig 1(c) (blue line)). We then conducted a "twin-experiment" to infer model parameters back from the model data (Fig 1(c)) and validate the ability of nonlinear optimization to recover the true parameter solution. The parameters were estimated using an interior point line parameter search algorithm [28] which was used earlier to build predictive neuron models [1, 2, 31]. The assimilation window had $n = 10, 000$ mesh points. The mesh size was $\Delta t = 20\mu$s ($T = 200$ms). All 41 parameters of the optimal solution $p_0^*$ are listed in Table 2. Each parameter estimate was found to be within 0.2% of its true value.

We then synthesized experimental data by adding noise to the useful membrane voltage: $V_{\exp}(t) = V_{use}(t) + \epsilon_{\sigma\zeta}(t)$. We generated $R = 1000$ different time series with different noise realizations $\zeta$ to generate a statistical distribution of estimated parameters $\pi(p_{\sigma\zeta}^*)$. Convergence to the optimum solution was secured by initializing the parameter search at $p_0^*$.

Fig 2(a) shows the distribution of estimated parameters centred on their mean value ($\sigma = 0.75$mV). The sloppiest parameters are characteristically the recovery time constants, and more specifically those of the Na channel ($t_m$), HCN channel ($t_z$), and low threshold Ca$^{2+}$ channel ($t_q$). The effect of increasing noise amplitude from $\sigma = 0$ to 0.75mV is to broaden the distribution of estimated parameters. This is shown in Fig 2(b) and 2(c) for the HCN recovery time ($t_z$) and the maximum Calcium permeability ($p_T$). As noise increases from $\sigma = 0$ to 0.75mV the MLE of parameter $t_z$ remains approximately constant and the standard deviation broadens symmetrically. In contrast, the MLE of parameter $p_T$ increases monotonically as noise increases from $\sigma = 0$ to 0.75mV. The parameter distribution is asymmetrical even at low noise amplitude.

**Table 1. Ion channels of the RVLM neuron.** Current densities with maximal conductances $g_\alpha$, $\alpha \in \{NaT, K, HCN, L\}$; sodium and potassium reversal potentials, $E_{Na}$ and $E_K$; hyperpolarized-activated cation reversal potential $E_{HCN}$ = -43mV [69]; leakage potential $E_L$ [70]. $m$ and $h$ are the state variables of the activation and inactivation gates of the NaT channel. $n$ is the activation gate of potassium. $z$ is the HCN activation gate. The Calcium current is given by the Goldman-Hodgkin-Katz equation Eq 13 [71].

| ID | Channel | Current density | Maximal conductance |
|----|---------|-----------------|---------------------|
| NaT | Fast and transient Na$^+$ current | $J_{NaT} = g_{NaT}\, m^3 h(E_{Na} - V)$ | $g_{NaT} = 110\text{mS.cm}^{-2}$ |
| K | Transient depolarization activated K$^+$ current | $J_{K1} = g_K\, n^4(E_K - V)$ | $g_{K1} = 5\text{mS.cm}^{-2}$ |
| HCN | Hyperpolarization-activated cation current | $J_{HCN} = g_{HCN}z(E_{HCN} - V)$ | $g_{HCN} = 0.092\text{mS.cm}^{-2}$ |
| CaT | Low threshold Ca$^{2+}$ current | $J_{CaT} = GHK$ | - |
| L | Leakage channels | $J_L = g_L(E_L - V)$ | $g_L = 0.066\text{mS.cm}^{-2}$ |

We then used the 1000 parameter estimations to compute the PDFs and reveal the effects of model nonlinearity. The PDFs of the parameters representing the transition regions of the activation curves of K$^+$ ($\delta V_n$) and HCN ($\delta V_z$) are plotted in Fig 2(d) and 2(e) respectively. These PDFs are compared to their Gaussian best fit (solid red line) at three noise amplitudes, $\sigma =$ 0.25, 0.5, 0.75 mV. As observed for $t_z$, the MLE of parameter $\delta V_n$ is independent of noise, the PDF remains approximately Gaussian at all noise amplitudes, and its standard deviation increases as noise amplitude increases (Fig 2(d)). In contrast, $\delta V_z$, like $p_T$ above, has a non-Gaussian PDF, and its MLE shifts to a lower voltage as $\sigma$ increases (Fig 2(e)).

Lastly, we investigated the correlations between estimated parameters and investigated the effect of increasing noise amplitude on parameter correlations. For this we calculated the covariance matrix:

$$\hat{\Sigma}_{l,m} = \frac{1}{R - 1} \sum_{r=1}^{R} \left(p_{l,r} - \bar{p}_l\right)\left(p_{m,r} - \bar{p}_m\right), \tag{9}$$

which is related to the Hessian through $\hat{H} = \hat{\Sigma}^{-1}$. $R$ is the number of noise realizations and hence the statistical sample of parameter sets used to calculate the covariance matrix. We calculated the eigenvalues $\lambda_k^2$ of $\Sigma$ which are the squares of the principal half-lengths of the data misfit ellipsoid (Fig 2(f)). Clearly the RVLM model parameters exhibit correlations spanning several orders of magnitude. Most parameters are well-constrained. However not all correlations vanish as $\sigma \to 0$. The two leftmost points (black circles) indicate pairs of parameters which remain correlated irrespective of noise amplitude. These parameters are the recovery time constants $t_m$, $t_z$ and $t_q$ already noted in Fig 2(a) to have a wider dispersion than the other parameters. Unsurprisingly, increasing noise amplitude increases parameter correlations. We also calculated the dependence of the standard deviation of the PDF, $\sigma_p$, as a function of the noise amplitude $\sigma$ (Fig 2(g)) for arbitrarily chosen parameters. Note the sub-linear dependence tending to saturation.

## Regularization of convergence by additive noise

Due to the nature of data assimilation, certain initial guesses of state variables and parameters may lead to sub-optimal solutions which are local minima of the data misfit function. The local minimum nearest to the global minimum was identified by running parameter searches initialized at random points in parameter space. This local minimum in the absence of additive noise is given in Table 2 as $\boldsymbol{p}_0^\ell$. We now switch on noise and study the effect of noise amplitude $\sigma$ and noise realization $\zeta$ on the relative positions of $\boldsymbol{p}_{\sigma\zeta}^*$ and $\boldsymbol{p}_{\sigma\zeta}^\ell$.

**Table 2. Parameters of the RVLM neuron model.** From left column to right column: parameter search interval, $[\boldsymbol{p}_L, \boldsymbol{p}_U]$; true parameters used to synthesize model data, $\boldsymbol{p}_{true}$; optimal parameters estimated at the true global minimum of the cost function, $\boldsymbol{p}_0^*$ ($\sigma = 0$); sub-optimal parameters estimated at the global minimum shifted by noise, $\boldsymbol{p}_{\sigma\zeta}^*$ ($\sigma = 0.5$mV); sub-optimal parameters estimated at the local minimum, $\boldsymbol{p}_0^\ell$ ($\sigma = 0$), nearest to the global minimum $\boldsymbol{p}_0^*$.

| Ion | Parameter | | Data | | Estimates | | |
|---|---|---|---|---|---|---|---|
| | | | $\boldsymbol{p}_L, \boldsymbol{p}_U$ | $\boldsymbol{p}_{true}$ | $\boldsymbol{p}_0^*$ | $\boldsymbol{p}_{\sigma\zeta}^*$ | $\boldsymbol{p}_0^\ell$ |
| | $C$ | $\mu$F.cm$^{-2}$ | 1.0, 1.0 | 1.0 | 1.0 | 1.0 | 1.0 |
| | $E_{\mathrm{Na}}$ | mV | 42, 50 | 41 | 41.007 | 41.075 | 60.000 |
| | $E_{\mathrm{K}}$ | mV | -90, -80 | -100 | -100.005 | -100.763 | -90.000 |
| | $E_{\mathrm{H}}$ | mV | -30, -5 | -43 | -42.963 | -42.793 | -30.000 |
| | $E_{\mathrm{Leak}}$ | mV | -110, -65 | -65 | -64.999 | -64.964 | -66.541 |
| | $A$ | $\times 10^4\,\mu$m$^2$ | $20^2$—$50^2$ | 2.90 | 2.90 | 2.91 | 2.90 |
| NaT | $g_{\mathrm{NaT}}$ | mS.cm$^{-2}$ | 100, 120 | 69 | 68.912 | 69.924 | 100.000 |
| $m$ | $V_m$ | mV | -49, -27 | -39.92 | -39.921 | -39.965 | -30.931 |
| | $\delta V_m$ | mV | 5, 32 | 10 | 10.000 | 9.949 | 15.850 |
| | $\delta V_{\tau m}$ | mV | 5, 23.39 | 23.39 | 23.380 | 23.254 | 0.100 |
| | $t_m$ | ms | 0.02, 0.7 | 0.143 | 0.143 | 0.157 | 0.815 |
| | $\varepsilon_m$ | ms | 0.012, 7 | 1.099 | 1.099 | 1.094 | 19.543 |
| $h$ | $V_h$ | mV | -79, -39 | -65.37 | -65.365 | -65.558 | -52.863 |
| | $\delta V_h$ | mV | -35, -5 | -17.65 | -17.652 | -17.629 | -13.752 |
| | $\delta V_{\tau h}$ | mV | 4, 43 | 27.22 | 27.218 | 27.670 | 14.107 |
| | $t_h$ | ms | 0.02, 90 | 0.701 | 0.701 | 0.684 | 0.502 |
| | $\varepsilon_h$ | ms | 1, 470 | 12.9 | 12.898 | 12.942 | 10.629 |
| K | $g_K$ | mS.cm$^{-2}$ | 0 | 6.9 | 6.905 | 6.736 | 2.232 |
| $n$ | $V_n$ | mV | -69, -21 | -34.58 | -34.557 | -34.763 | -39.654 |
| | $\delta V_n$ | mV | 5, 34 | 22.17 | 22.178 | 21.932 | 13.118 |
| | $\delta V_{\tau n}$ | mV | 5, 34 | 23.58 | 23.588 | 23.851 | 21.556 |
| | $t_n$ | ms | 0.01, 5.4 | 1.291 | 1.291 | 1.273 | 0.434 |
| | $\varepsilon_n$ | ms | 0.002, 23 | 4.314 | 4.311 | 4.248 | 6.416 |
| CaT | $p_{\mathrm{T}}$ | $\times 10^{-4}$ cm.s$^{-1}$ | 0, 80 | 1.035 | 1.035 | 0.210 | 0.130 |
| | $V_q$ | mV | -80, -35 | -65.5 | -65.491 | -64.483 | -67.767 |
| $q$ | $dV_q$ | mV | 5, 39 | 12.4 | 12.391 | 14.003 | 9.958 |
| | $\delta V_{\tau q}$ | mV | 10, 57 | 27 | 27.123 | 28.911 | 14.985 |
| | $t_q$ | ms | 0.02, 0.9 | 0.719 | 0.693 | 2.232 | 7.556 |
| | $\varepsilon_q$ | ms | 0.5, 97 | 13.05 | 13.059 | 11.759 | 8.370 |
| $r$ | $V_r$ | mV | -90, -55 | -86 | -86.011 | -73.916 | -74.356 |
| | $\delta V_r$ | mV | -34, -5 | -8.06 | -8.065 | -4.547 | -3.962 |
| | $\delta V_{\tau r}$ | ms | 3, 55 | 16.71 | 16.760 | 9.829 | 0.100 |
| | $t_r$ | ms | 5, 190 | 28.17 | 28.120 | 27.435 | 55.095 |
| | $\varepsilon_r$ | mV | 0.5, 7000 | 288.68 | 287.067 | 319.355 | 1000.000 |
| HCN | $g_{\mathrm{H}}$ | mS.cm$^{-2}$ | 0, 10 | 0.150 | 0.150 | 0.149 | 0.177 |
| $z$ | $V_z$ | mV | -90, -40 | -76 | -76.001 | -76.297 | -79.121 |
| | $\delta V_z$ | mV | -30, -5 | -5.5 | -5.517 | -5.430 | -11.876 |
| | $\delta V_{\tau z}$ | mV | 5, 40 | 20.27 | 20.273 | 21.861 | 100.000 |
| | $t_z$ | ms | 0.1, 500 | 6.31 | 6.348 | 0.100 | 10.000 |
| | $\varepsilon_z$ | mV | 0.1, 5000 | 55.05 | 55.019 | 60.471 | 50.323 |
| Leak | $g_L$ | mS.cm$^{-2}$ | 0.01, 0.6 | 0.465 | 0.465 | 0.463 | 0.482 |

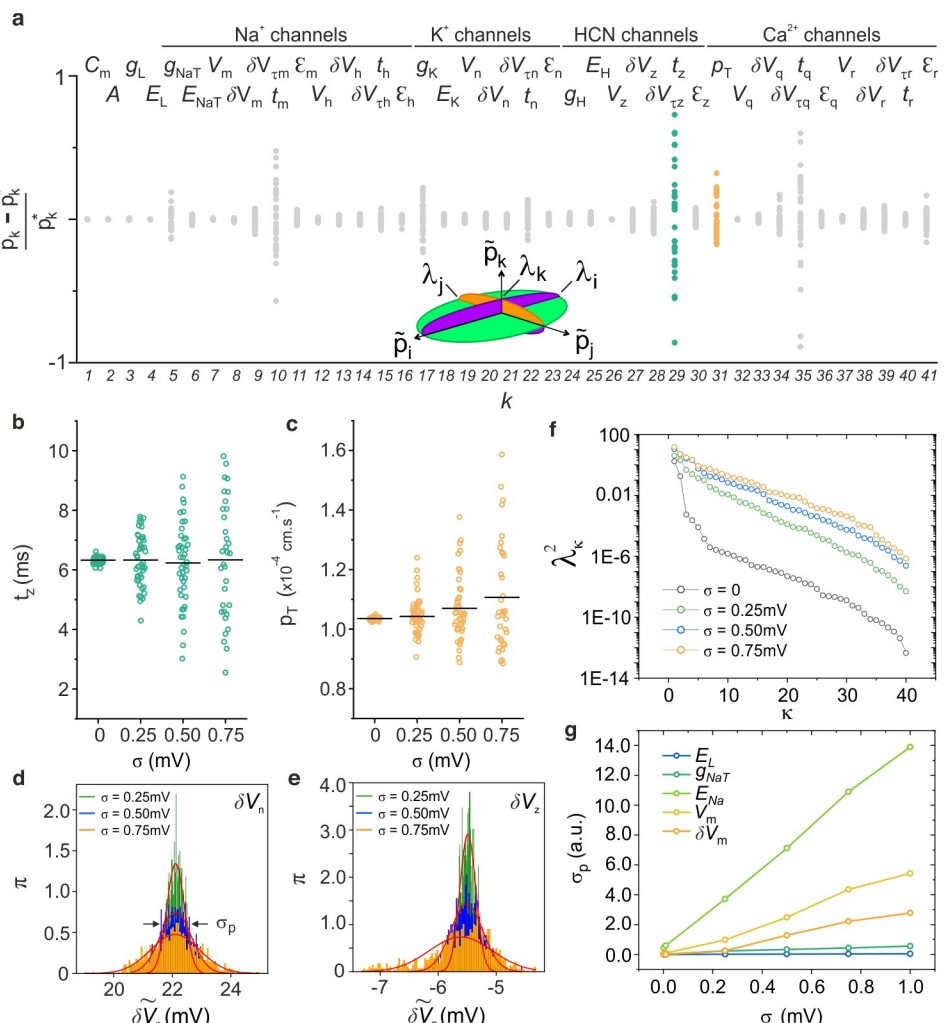

**Fig 2. Probability distribution of estimated parameters. (a)** Scatter plot of parameters $p_k$, $k = 1\ldots41$, estimated by assimilating the RVLM membrane voltage incorporating different realizations of Gaussian noise. Noise amplitude: $\sigma = 0.75$mV. The dependence of this distribution on noise amplitude is plotted for 2 parameters: **(b)** the recovery time $t_z$ of HCN inactivation gate and, **(c)** the maximum permeability of the CaT ion channel, $\bar{p}_{CaT}$. **(d,e)** Probability density functions (PDF) of parameters $t_z$ and $\bar{p}_{CaT}$ calculated at increasing noise amplitudes $\sigma = 0.25$, 0.50 and 0.75mV. Statistical sample: 1000 parameter sets extracted for different noise realizations. The initial condition was $\boldsymbol{p}_0^*$ **(f)** Eigenvalue spectrum of the 41 × 41 covariance matrix of parameter estimates. The $\lambda_\kappa$, $\kappa = 1\ldots41$ are the semi-axes of the data misfit ellipsoid $\delta c = f(\sigma, \zeta)$ and the $\lambda_k^2$ are the eigenvalues of covariance matrix $\Sigma$. Spectra are calculated at four noise amplitudes: $\sigma = 0$, 0.25, 0.50 and 0.75mV. **(g)** Relationship between the standard deviation of a parameter, $\sigma_p$, and the noise amplitude, $\sigma$.

Our regularization method is depicted schematically in Fig 3(a). This relies on the noise-induced shift in parameter solutions. We begin by choosing one realization of additive noise ($\zeta$) before varying the noise amplitude in the range $-0.5$mV $< \sigma < +0.5$mV. A negative value of $\sigma$ here implies a temporal realization of noise with negative amplitude but same Gaussian probability distribution. (i) Starting from $\sigma = 0$, the local and global minima, $\boldsymbol{p}_0^\ell$ (pink star) and $\boldsymbol{p}_0^*$ (red star), are separated by a saddle point in the cost function surface (open dot). (ii) As $\sigma$ increases, the local and global minima shift relative to one another, getting closer or further apart depending on the sign of $\sigma$. When $\boldsymbol{p}_{\sigma\zeta}^*$ and $\boldsymbol{p}_{\sigma\zeta}^\ell$ (blue dots) approach one another, there

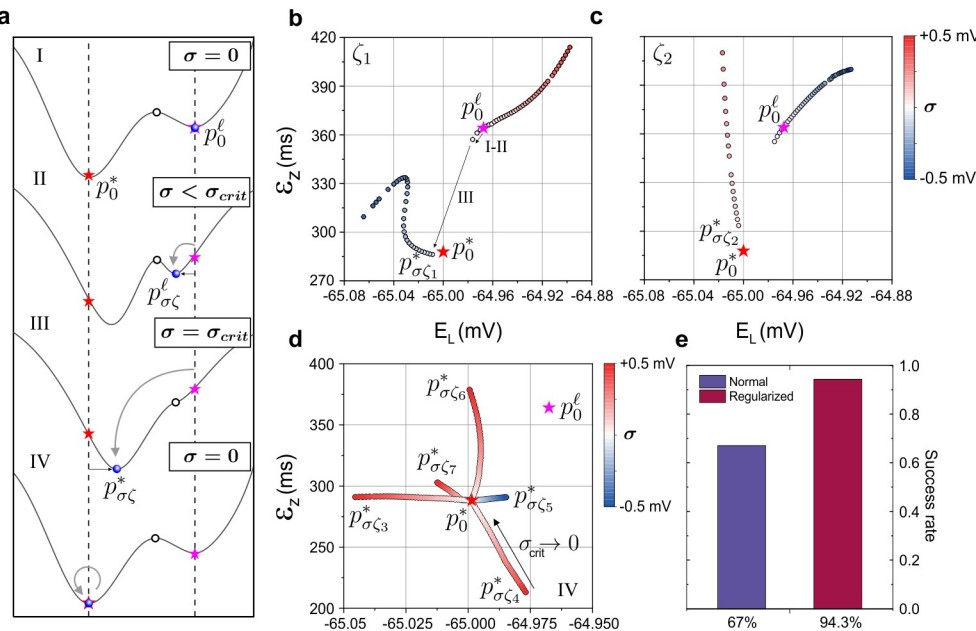

**Fig 3. Regularization of parameter search. (a)** Profile of the data error misfit function $\delta c$ plotted along a straight line passing through the global minimum $p_0^*$ (red star) and the nearest local minimum $p_0^\ell$ (magenta star). (I) In the absence of noise ($\sigma = 0$), a saddle point separates $p_0^\ell$ and $p_0^*$ (open dot). (II) Increasing noise amplitude up to a critical value $\sigma < \sigma_{crit}$ shifts the local solution, $\boldsymbol{p}_0^\ell \to \boldsymbol{p}_{\sigma\zeta}^\ell$, and the global solution, $\boldsymbol{p}_0^* \to \boldsymbol{p}_{\sigma\zeta}^*$ (blue dots). (III) At $\sigma_{crit}$, the barrier at the saddle point vanishes. Hence, the local minimum $\boldsymbol{p}_{\sigma_{crit}\zeta}^\ell$ merges with the saddle point. (IV) Parameter search initialized at $\boldsymbol{p}_{\sigma_{crit}\zeta}^*$ converges smoothly to the optimal solution $p_0^*$ as noise vanishes. In this way, parameter search is regularized. **(b)** Trajectory of the local solution parametrized by noise as the noise amplitude varies from $\sigma = -0.5$mV to $+0.5$mV. The noise amplitude is colour coded in each dot. The noise realization remains the same ($\zeta_1$). The 41-dimensional trajectory is projected onto the 2D plane ($E_L, \varepsilon_z$). At $\sigma_{crit} = -40\mu$V, $\boldsymbol{p}_{\sigma\zeta}^\ell$ merges with $\boldsymbol{p}_{\sigma\zeta}^*$ (step III). **(c)** Same as in (b) but for a trajectory calculated with a different noise realization, $\zeta_2$. Here $\sigma_{crit} = +50\mu$V. **(d)** Various trajectories of the solution $\boldsymbol{p}_{\sigma\zeta}^*$ during step IV. The different starting points are the shifts induced by different realizations of noise, $\zeta_3, \ldots, \zeta_8$. **(e)** Probability of convergence to the optimal solution with noise regularization (red) and without (blue). The success rate was calculated from a statistical sample of 150 parameter solutions computed from random parameter initializations.

exists a critical noise amplitude $\sigma_{crit}$ (iii) where the saddle point and the local minimum merge inducing a saddle-node bifurcation [49] towards the global minimum: $\boldsymbol{p}_{\sigma\zeta}^\ell \to \boldsymbol{p}_{\sigma\zeta}^*$. (iv) $\boldsymbol{p}_{\sigma\zeta}^*(\zeta)$ is then set as the new initial guess of the parameter search. $\sigma$ is then ramped down to zero from $\sigma_{crit}$ to obtain the optimal parameter solution $\boldsymbol{p}_0^*$.

Steps (i) to (iii) are demonstrated numerically in Fig 3(b) and 3(c). The parameter search was initialized at the local minimum $\boldsymbol{p}_0^\ell$ where the cost function was $c(\boldsymbol{x}(0), \boldsymbol{p}_0^\ell) = 9.105306 \times 10^{-5}$. In contrast, the cost function at the global minimum $\boldsymbol{p}_0^*$ was almost two orders of magnitude lower at $C(\boldsymbol{x}(0), \boldsymbol{p}_0^*) = 1.179402 \times 10^{-6}$. The state variables were initialized at the same values throughout. The data time series had $n = 10,000$ points and $\Delta t = 20\mu$s. Two different noise realizations $\zeta_1$ and $\zeta_2$ were applied in Fig 3(b) and 3(c) respectively. Initializing the estimation procedure at $\boldsymbol{p}_0^\ell$, the parameter solution was calculated and projected in the two-dimensional plane ($\varepsilon_z, E_L$) as $\sigma$ varied from 0 to $+0.5$ (red dots) and 0 to $-0.5$ (blue dots). $\varepsilon_z$ is a parameter of the HCN activation gate which gives the difference in recovery times between the half-open and fully open state of the gate. $E_L$ is the leak reversal potential. The same qualitative results are observed in other projection planes involving different pairs of parameters

in Table 2. At $\sigma = 0$, the parameter solution remains the local minimum (Fig 3(b) and 3(c), magenta star). For $\sigma > 0$, the local and global minima move away from one another causing $\boldsymbol{p}_{\sigma\zeta}^{\ell}$ to shift monotonically away from $\boldsymbol{p}_{\sigma\zeta}^{0}$ as $\sigma$ increases (red dots). In contrast, when $\sigma < 0$, the distance between the local and global minima decreases. At $\sigma_{crit} = -40\mu$V, the saddle point vanishes followed by an abrupt transition from the local minimum $\boldsymbol{p}_{\sigma\zeta}^{\ell}$ to the global minimum $\boldsymbol{p}_{\sigma\zeta}^{*}$. The effect of using a different noise realization $\zeta_2$ in Fig 3(c) is to change the path of the solution in parameter space. The saddle-node bifurcation also occurs at a different noise amplitude of $\sigma_{crit} = +50\mu$V.

Steps (iii) to (iv) are demonstrated in Fig 3(d). The optimal solution $\boldsymbol{p}_0^*$ was recovered by ramping down $\sigma$ from $\sigma_{crit}$. The trajectories of $\boldsymbol{p}_{\sigma\zeta}^*$ converge to $\boldsymbol{p}_0^*$ as $\sigma$ is progressively decreased from $\sigma_{crit}$. Fig 3(d) shows the trajectories calculated for 5 different noise realizations $\zeta_1 \dots \zeta_5$. Fig 3(d) thus demonstrates the dependence of the noise-induced parameter offset on noise realization, as predicted by Eq 7.

Therefore, the two-step procedure we have described is useful to regularize convergence towards the global minimum. The algorithm of the regularization method may be summarized as follows: (i) Solve the inverse problem using smooth data. The solution may be optimal or sub-optimal. (ii) Apply additive noise to the data and vary its amplitude while keeping its realisation constant until an abrupt step in both $\delta\boldsymbol{p}$ and $\delta c$ is observed. (iii) Progressively reduce noise amplitude to zero to obtain the optimal parameter solution. Assimilations of the RVLM neuron model starting from 150 random initial guesses of parameters and state variables were found to converge to the optimum solution with a probability of 94.3% using noise regularization, and 67% without. In the other 5.7% and 33% of cases, convergence terminated at local minima. (Fig 3(e)).

## Decorrelating parameters

Parameter uncertainty and correlations may arise from incomplete fulfilment of identifiability conditions if the stimulation protocol is ill-chosen. For conductance models, this means that the assimilation window must contain multiple action potentials as most model parameters control the dynamics of depolarization. In addition, current protocols must include (i) current steps of different durations to probe the recovery of ionic gates with different kinetics, and (ii) current steps of different amplitude to extract information from the depolarized, sub-threshold and hyperpolarized states of a neuron. These complex current protocols are required to decorrelate the model constraints (Eq 2) linearized at consecutive time points of the assimilation window. Increasing the window length also contributes to better constrained global parameter solutions. The drawback, however, is that as $n$ increases beyond $n_{max} \approx 10^4$ points, the cost function becomes highly irregular due to an increased number of local minima [50, 51]. In order to increase the length $T$ of the assimilation window while keeping $n < n_{max}$, we introduce a smart sampling method which samples sub-threshold oscillations with a larger step size than action potentials. For membrane voltages above -65mV, we apply a mesh size of $\Delta t_1 = 10\mu$s whereas sub-threshold oscillations are sampled with a mesh size $\Delta t_2 = n\Delta t_1$ (Fig 4(a)). The rationale for this is that sub-threshold oscillations are controlled by fewer parameters than the depolarized state. Since time intervals of membrane depolarization are few and far apart, this approach allows considerable increases in duration of the assimilation window while keeping $n$ constant (see Methods).

We first studied the effect of the length of the assimilation window on parameter correlations by computing the spectrum of eigenvalues of the covariance matrix $\hat{\Sigma}$ (Fig 4(b)). The covariance matrix was generated by assimilating model membrane voltages with $R = 1000$ different realizations of additive noise of amplitude $\sigma = 0.75$mV. The assimilation window

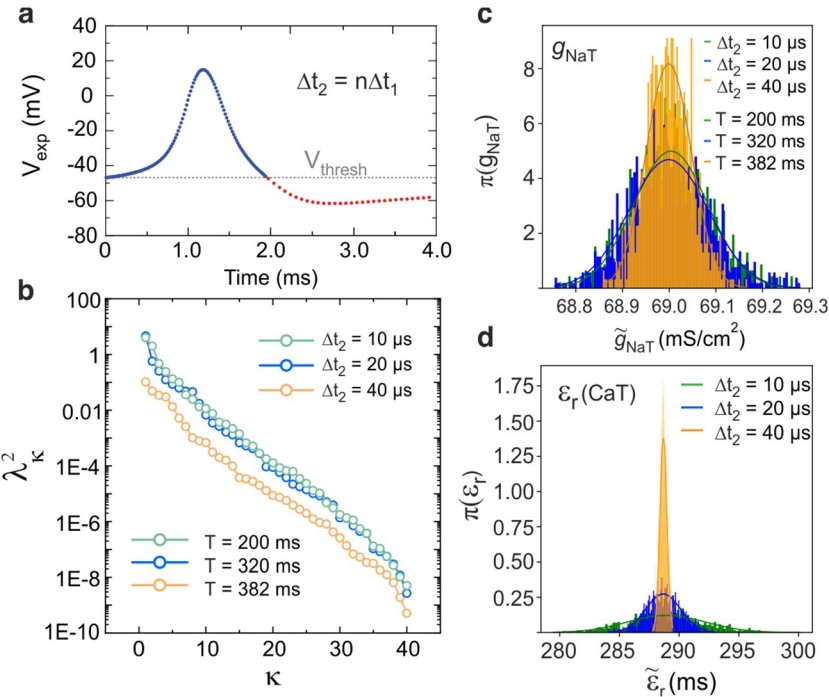

**Fig 4. Increasing the duration of the assimilation window reduces parameter uncertainty. (a)** An adaptive step size was used to increase the duration of the assimilation window while keeping the size the problem constant and equal to $n = 10,000$ samples. The step size was $\Delta t_1 = 0.01$ms during the depolarization time intervals ($V_{exp} > -63$mV) and $\Delta t_2 = m\Delta t_1$, $m = 1, 2, 4$, elsewhere. **(b)** Dependence of the parameter correlations as the duration of the assimilation window increase from $T = 200$ms ($m = 1$), 320ms ($m = 2$) to 382ms ($m = 4$). The eigenvalues of the covariance matrix were calculated from parameters estimated from randomly initialized parameters and state variables. Additive noise had amplitude $\sigma = 0.25$ mV. Posterior distribution function of two parameters chosen for controlling **(c)** *action potentials* via the sodium conductance $g_{NaT}$ and **(d)** *sub-threshold oscillations* via calcium kinetics $\varepsilon_r$. Statistical sample for histograms (b,d): 1000 assimilations started at the global minimum with a unique noise realization.

had 10, 001 data points but their time intervals varied. The spectrum of eigenvalues is plotted for increasingly wide assimilation windows corresponding to $\Delta t_1 = 10\mu$s ($T = 200$ms), $20\mu$s ($T = 320$ms), $40\mu$s ($T = 382$ms). Fig 4(b) shows that increasing the duration of the assimilation window *uniformly* reduces correlations, $\lambda_k^2$, for *all* 41 parameters. Compare this with Fig 2(f) where some parameters remain highly correlated even at $\sigma \to 0$. Fig 4(c) and 4(d) show the progressive narrowing of the PDF of the $g_{NaT}$ and $\varepsilon_{CaT}$ parameters as $T$ increases. Conductances such as $g_{NaT}$ are already well constrained hence their PDF becomes marginally narrower as $T$ increases. In contrast, the standard deviation of loosely constrained recovery time constants in Fig 2(a) decrease by an order of magnitude as the duration of the assimilation window increases from $T = 200$ms to 382ms (Fig 4(d)). We have therefore shown that long assimilation windows increase parameter identifiability and considerably reduce sloppiness.

## Comparing model predictions with local and global parameters

We finally compare the predictions of models configured with 3 sets of parameters: $\boldsymbol{p}_0^*$, $\boldsymbol{p}_0^\ell$ and $\boldsymbol{p}_{\sigma\zeta}^*$, a vicinal location to the global minimum defined as the global minimum shifted by noise. These parameters are listed in Table 2. Fig 5(a) shows the locations of $\boldsymbol{p}_0^\ell$ (purple dot) and $\boldsymbol{p}_{\sigma\zeta}^*$ (orange dot) on the data misfit surface relative to $\boldsymbol{p}_0^*$ (red dot). The Euclidean norm was used

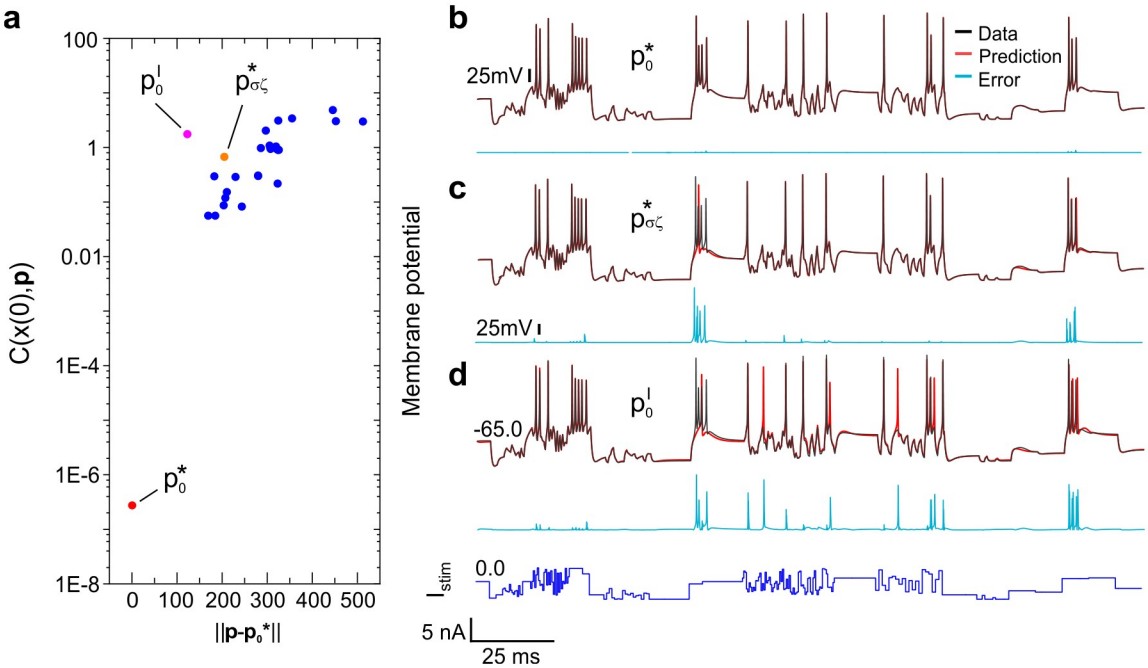

**Fig 5. Effect of optimal and sub-optimal parameters on model predictions. (a)** Value of the cost function at the site of local minima (purple/orange/blue dots) in the vicinity of the global minimum (red dot) plotted as a function of the distance to the global minimum defined by the Euclidean metric. The blue dots are the local minima situated further away from the global minimum. **(b-d)** Reference membrane voltage (black line) induced by the current protocol (dark blue line). The membrane voltage predicted by configuring the RVLM model with parameters: **(a)** $p_0^*$, **(b)** $p_{\sigma\zeta}^*$, **(c)** $p_0^\ell$ is shown as the red line. The difference between the predicted voltage and the reference voltage is the prediction error (cyan lines).

to evaluate the distance in parameter space $\|p - p_0^*\|$ to the optimum solution. We show here that predictions made with sub-optimal parameters $p_{\sigma\zeta}^*$, $p_0^\ell$ are always discernible from those made with the optimal set $p_0^*$.

The predictions of the three RVLM neuron models configured with parameters $p_0^*$, $p_{\sigma\zeta}^*$ and $p_0^\ell$ are shown in Fig 5(b), 5(c) and 5(d) respectively (red lines). These are compared to the model data synthesized using $p_{true}$ (black line). The prediction error is the cyan line (Fig 5(b)–5(d)). Predictions obtained with $p_0^*$ are identical to the model data. Interestingly, prediction accuracy is maintained in spite of residual numerical error in $p_0^*$. These computational errors do not diminish the predictive power of the model (Fig 5(b)). In contrast, predictions made by configuring the RVLM model with $p_{\sigma\zeta}^*$ show systematic discrepancies at the site of action potentials (Fig 5(c)). Spike bursts are completely missed and the height of action potentials is incorrect. The sub-threshold dynamics is, however, represented with great accuracy. Similarly, predictions made with $p_0^\ell$ show some missing spikes and some additional ones (Fig 5(d)). These results suggest that the original parameters form the one and only set capable of predicting the experimental time series. Hence, the injected current is sufficiently discriminating for the identifiability condition to be validated. The membrane voltage time series encodes the single-valued parameter solution as prescribed by Takens' theorem. We have further verified in S1 Fig that a current protocol consisting of long rectangular steps fail to constrain all model parameters. This demonstrates the importance of selecting external stimuli that probe the full dynamic range of the nonlinear system for parameters to be identifiable.

## Discussion

The significance of parameter estimation methods for extracting information from biological systems has recently been discussed [6, 9]. An increasingly prevalent view among biologists is that parameters estimated from biological models are universally sloppy [7] and that disparate sets of parameters can generate identical neuronal oscillations [37, 52]. The notion that biocircuits must incorporate functional overlap is consistent with the observation of brain remodelling and ageing. For example, the brains of the elderly lose between 2% and 4% of their peak number of neurons without significant decrease in cognitive abilities [53]. Therefore, if the function of a biological system is underpinned by redundant degrees of freedom, can one reasonably expect to infer its internal structure from observations of its dynamics?

The answer from nonlinear science is that the parameters and initial conditions that control neuronal oscillations can generally be inferred from the observation of its membrane voltage over a finite time interval [22, 23, 27]. However, there are conditions to satisfy. The condition of *observability* is satisfied by choosing a number of data points greater than $L + K$. This condition is easily met. Both Toth et al [31] and ourselves in Table 2 have demonstrated the system is observable by recovering the original parameters in twin experiments. The second condition—*identifiability*—requires the system to be driven by an external stimulus with the appropriate range of dynamics and current amplitudes to constrain all parameters. For example, parameters extracted from data acquired under simpler current injection (S1 Fig) are not identifiable and are poorly constrained in contrast to those listed in Table 2 ($\boldsymbol{p}_0^*$). A driving force with complex dynamics is therefore necessary to warrant identifiability. In addition, increasing the duration of the assimilation window matters to reduce correlations between parameters and increase identifiability as observed by others [37, 50]. We have achieved this in Fig 4 by introducing an adaptive step size within our gradient descent algorithm. A second advantage of using an adaptive step size is that it allows longer assimilations windows and longer current steps to be applied (500ms). This is essential to quantify the effect of slow decaying currents on the long term potentiation of neurons [54]. When the conditions of observability and identifiability are met, we have shown in Fig 5 that sub-optimal parameters (at local minima) always give sub-optimal predictions which are easily distinguished from predictions by the optimal set of parameters. Therefore, under these conditions single-valued parameter solutions may be obtained from the time series observations of the neuron membrane voltage.

One more complication is the presence of local minima in the cost function. The global minimum becomes harder to distinguish from local minima when the noise-induced error in the cost function becomes comparable to the data misfit error at a local minimum. In Fig 3, we introduce a regularization method which makes constructive use of additive noise to bias the gradient descent algorithm towards the global minimum when it would otherwise remain stuck in a local minimum. This method is well suited to the assimilation of actual neuron data acquired by low noise amplifiers in well-controlled experimental preparations for which experimental error remains a perturbation of the useful signal [1, 2]. The assimilation of very noisy data may still be approached using statistical inference methods such as expectation maximization frameworks [37, 38] or path integral methods [55]. However these methods rely on prior knowledge of parameter distribution functions whereas the present variational approach does not.

Modern data assimilation [34, 44, 56] introduces experimental and model error in the form of covariance products which weight each measurement with the error of the measuring apparatus. These approaches are not suitable for highly nonlinear systems where a Gaussian shaped

probability density on data does not translate into a Gaussian shaped probability density on parameters. Moreover the same electrophysiological apparatus is used to record all data points in the time series. Given each measurement carries the same error, this approach is fact reduces to our least-squares cost function (Eq 2). The nonlinearity of the conductance model implies that Bayesian approaches are no longer applicable to estimating MLE and standard deviation of parameter PDFs [4, 5, 26, 32, 57–61]. Our work has studied separately the effect of experimental and model error. We found that both errors shift the parameter solution on the data misfit surface. However, the primary cause of the parameter offset is experimental error with a second order contribution from model error. Our results identify the interplay between model nonlinearity and the realization of noise across the assimilation window as the reason for the parameter offset and its dependence on noise realization. An important consequence of this noise-induced shift is that the parameter solution inferred in the presence of experimental error is invariably wrong.

Our results show that while biocircuits may exhibit functional overlap in their parameters, their underlying configuration can still be inferred provided an external driving force is applied. Parameter identifiability is always relative to the degree of sophistication of external stimulation. Unsurprisingly, functional overlap between parameters is primarily observed in self-sustaining oscillators such as central pattern generators operating in the steady-state without external input [9, 10, 40]. For such systems, parameter overlap [6, 9] may be useful to compensate for loss of functionality [11], and parameter sloppiness may be pervasive [7]. However, recent experiments have shown that among all network configurations with apparent overlap, only a small subset of these was able to explain the adaptation of rhythmic outputs to temperature changes [62], and changes in pH levels [63]. There is no doubt that subjecting central pattern generators to a wider range of entrainments would further reduce the set of parameters compatible with the observed outputs, up to the point where a unique parameter solution would remain that characterises all electrical properties. There is therefore no theoretical limitation to inferring the underlying structure of ion channels or connectivity of small networks other than the ingenuity in designing stimulation protocols that fulfill identifiability conditions. Translated to the brain, redundancy may allow normal operation to continue with ageing but our work suggest that flexibility to adapt to external stimulation will decrease together with the size of its parameter space.

In conclusion, parameter redundancy and compensation is relative to external stimulation. Long and dynamically complex stimulation protocols were shown to reduce correlation between estimated parameters. We also quantified the effects of noise and model error and made constructive use of the induced parameter offset to increase the probability of convergence to the optimal set of parameters.

## Methods

### Conductance model

We model the parasympathetic neuron of the rostral ventrolateral medulla (RVLM). The RVLM neurons play a key role in cardiac regulation by accelerating heart rate and increasing the force of contraction of the heart muscle. In this way, these neurons compensate the action of vagal tone which reduces heart rate [47]. RVLM neurons have a greater complement of ion channels than the textbook Hodgkin-Huxley neuron [31]. This makes these neurons a good choice for evaluating the accuracy of the parameter estimation method when building models of actual neurons. The ion channels of RVLM neurons include transient sodium (NaT), potassium (K), low threshold calcium (CaT) and the hyperpolarization-activated cation current

(HCN) [48]. The equation of motion for the membrane voltage is:

$$C\frac{dV(t)}{dt} = -J_{NaT} - J_K - J_{CaT} - J_{HCN} - J_L + I_{\text{inj}}(t)/A, \tag{10}$$

where $C$ is the membrane capacitance, $V$ is the membrane potential, $I_{inj}(t)$ is the injected current protocol, $A$ is the neuron surface area, and $J_{ion}$ are the voltage-dependent ionic current densities across the cell membrane. The equations of individual ionic currents are given in Table 1. These currents depend on maximum ionic conductances ($g_{NaT}, g_K, g_{HCN}$), sodium, potassium and HCN reversal potentials ($E_{Na}, E_K, E_{HCN}$), and gate variables ($m, h, n, p, q, s$). The control term $u(t_n)[V_{\exp}(t_n) - V(t_n)]$ was added to the right hand side of Eq 10 to eliminate the occurrence of positive conditional Lyapunov exponents and smooth convergence [64]. Ionic gates are assumed to recover from changes in membrane voltage according to a first order equation:

$$\frac{dx}{dt} = \frac{x_\infty(V(t)) - x(t)}{\tau_x(V(t))}, \tag{11}$$

where $x \in \{m, h, n, s\}$ represents the state of activation and inactivation of the NaT, K and HCN ionic gates (Table 1). The (in)activation curve of individual gates is modelled as:

$$x_\infty(V) = \frac{1}{2}\left(1 + \tanh\frac{V - V_{tx}}{\delta V_x}\right), \qquad \tau_x = t_x + \varepsilon_x\left(1 - \tanh^2\frac{V - V_{tx}}{\delta V_{\tau x}}\right), \tag{12}$$

where $V_{tx}$ is the (in)activation voltage threshold of the gate, $\delta V_x$ is the width to the transition region from closed to open states and, $\delta V_{\tau x}$ is the half-width-at-half-maximum of the bell-shaped voltage dependence of the recovery time. The recovery time is $t_x + \varepsilon_x$ at the opening threshold of the gate and $t_x$ in the depolarized and hyperpolarised states.

The transient low threshold calcium current is given by the Goldman-Hodgkin-Katz (GHK) equation:

$$J_{CaT} = \bar{p} \cdot p^2 \cdot q \cdot z^2 \cdot \frac{VF^2}{RT} \cdot \frac{[Ca^{2+}]_i - [Ca^{2+}]_o \cdot \exp\frac{-zFV}{RT}}{1 - \exp\left(\frac{-zFV}{RT}\right)}, \tag{13}$$

where $p$ and $q$ are the activation and inactivation variables of the CaT channel. $\bar{p}$ is the maximal permeability, $[Ca^{2+}]_i$ and $[Ca^{2+}]_o$ are the intra- and extracellular calcium concentrations, $z = 2$ is the valence of $Ca^{2+}$, $F$ is Faraday's constant, $R$ is the ideal gas constant, and $T = 298.15K$. The GHK equation was expanded about $V = 0$ into a Horner polynomial of order $n = 25$ to approximate Eq 12 over the range of the membrane voltages.

## Current protocols and model data

A set of current protocols $I_{inj}(t)$ consisting of current steps of different amplitudes and durations was synthesized to provide stimulation to the neurons (Fig 5, dark blue line). Each protocol was calibrated to induce depolarisation or hyperpolarisation over different time scales covering the recovery times of ion channels. Model data were synthesized by forward integration of these currents with the RVLM conductance model (Eqs 10–13) configured with the $\boldsymbol{p}_{true}$ set of parameters set in Table 2. The model equations were numerically integrated using the LSODA solver [65] which is able to resolve stiff and potentially unstable nonlinear systems [66]. Additive Gaussian noise $\epsilon_{\sigma\zeta}$ was generated with a pseudo random number generator and added to the model membrane voltage. In this way, we obtained both current and membrane voltage time series, $I_{inj}(t_i))$ and $V_{\exp}(t_i)$, used in data assimilation. The base sampling rate was 100kHz ($\Delta t = 10\mu s$).

## Nonlinear cost function optimization

The least-squares objective function constrained by model equations was minimized using interior point line parameter search [28]. The Lagrangian of the problem was constructed from the cost function, equality constraints and inequality constraints [29]. The Lagrangian was minimized under the Karush-Kuhn-Tucker conditions [30]. Equality constraints were obtained by linearizing the RVLM conductance model:

$$\frac{dx_l}{dt} = F_l(\boldsymbol{x}(t), \boldsymbol{p}, t), \quad l = 1 \cdots L, \tag{14}$$

at specific times across the assimilation window. The rate of change, $F_l()$, of state variable $l$ depends on all state variables $\boldsymbol{x}$, parameters $\boldsymbol{p}$ and time $t$. Inequality constraints were specified by the search intervals of individual parameters $p_{k,L} \leq p_k \leq p_{k,U}$, $k = 1 \ldots K$ which are listed in Table 2. The bounds of parameter search are the only user-specified inputs of the minimization problem. The Jacobian and Hessian matrices of the constraints and cost function were computed using symbolic differentiation (https://pypi.org/project/pydsi). Interior point optimization reformulates inequality constraints as logarithmic barriers whose height is reduced iteratively as the parameter search approaches the global minimum of the optimization surface [29]. Minimization was implemented iteratively using a Newton-type algorithm until first-order optimality conditions on the Lagrangian function $L(x)$ are met.

The equality constraints Eqs 10 and 11 were then discretized to connect the state variables evaluated at mesh points across the assimilation window. For this purpose mesh points were dynamically grouped according to the order of the interpolation formula and the variable step size, which we implemented to improve accuracy on parameter solutions. We linearized Eqs 10 and 11 according to Boole's interpolation which is accurate to $\mathcal{O}(\Delta t^7)$ [67] in contrast to Simpson rule's $\mathcal{O}(\Delta t^4)$ [31]:

$$\begin{aligned}
x_l(t_{i+4}) &= x_l(t_i) + 2\Delta t \left[ \frac{7}{25} F_l(\boldsymbol{x}(t_i)) + \frac{32}{45} F_l(\boldsymbol{x}(t_{i+1})) + \frac{12}{45} F_l(\boldsymbol{x}(t_{i+2})) \right. \\
&\qquad \left. + \frac{32}{45} F_l(\boldsymbol{x}(t_{i+3})) + \frac{7}{45} F_l(\boldsymbol{x}(t_{i+4})) \right].
\end{aligned} \tag{15}$$

Data points were grouped in sets of 5: $\{t_i, \ldots, t_{i+4}\}$. The state variable at $t_{i+4}$ was interpolated from evaluations of $F_l()$ at the 5 evenly spaced points separated by $\Delta t$. When the step size is constant, state variables are thus evaluated every $4\Delta t$.

We introduce an adaptive step size that samples sub-threshold oscillations with a lower resolution than action potentials. We therefore consider sub-threshold step sizes of $p\Delta t$ where $p = 2, 4. \ldots$ Our group of 5 points then spans a duration of $4p\Delta t$ within the adaptive step framework. The last point of one group is the same as the first point of the succeeding group. To warrant an integer number of groupings in the assimilation window, we chose $n$ to be an integer multiple of $4p$.

As Eq 16 constrains only one of the four points in the group, this condition alone does not force the solution to pass through the other 3 data points. The use of Eq 16 alone may support rapid oscillatory solutions which are undesirable [32]. In order to constrain the other 3 other points of the group, one needs to introduce additional Hermite conditions [31, 68]:

$$x_l(t_{i+1}) = \frac{1}{2}[x_l(t_i) + x_l(t_{i+2})] + \frac{\Delta t}{8}[F_l(t_i) - F_l(t_{i+2})], \tag{16}$$

$$x_l(t_{i+2}) \quad = \quad \frac{1}{2}[x_l(t_{i+1}) + x_l(t_{i+3})] + \frac{\Delta t}{8}[F_l(t_{i+1}) - F_l(t_{i+3})], \tag{17}$$

$$x_l(t_{i+3}) \quad = \quad \frac{1}{2}[x_l(t_{i+2}) + x_l(t_{i+4})] + \frac{\Delta t}{8}[F_l(t_{i+2}) - F_l(t_{i+4})]. \tag{18}$$

In practice, we find it is sufficient to evaluate only 2 out of 3 Hermite constraints to obtain smooth and accurate solutions. This reduces the computational effort without compromising accuracy on solutions.

The control variable $u$ and its time derivative $du/dt$ were bounded by $0 \leqslant u \leqslant 1\text{mV}$ and $-1\text{mV.ms}^{-1} < du/dt < +1\text{mV.ms}^{-1}$. The $u(t_i)$ were computed as an additional state variable across the assimilation window. To regularize convergence, we smoothed the fast oscillations of $u$ by applying the above Hermite conditions (Eq 18).

The adaptive step size was implemented automatically assigning step size $\Delta t$ during action potentials when $V_{\text{exp}} > -65\text{mV}$, and $p\Delta t$ ($p$ = 2 or 4) otherwise.

Assuming $G$ to be the number of data point groupings across the assimilation window, the problem overall had $L \times G$ constraints due to Boole's rule and $2(L+1) \times G$ constraints from Hermite's conditions.

## Supporting information

**S1 Fig. Dependence of parameter identifiability on the complexity of the current injection protocol.** (a) Dispersion of extracted parameters of the RVLM neuron model in response to a complex current stimulation protocol (grey line). (b) Same as (a) for a simpler current protocol (blue line). (c) Complex (grey) and simple (blue) current protocols used to stimulate the neuron and to constraint the parameters obtained in (a) and (b). (d) Size-ranked eigenvalue spectra of the covariance matrices $\hat{\Sigma}$ of parameters estimated using the two current protocols in (c).
(TIF)

## Author Contributions

**Conceptualization:** Alain Nogaret.

**Formal analysis:** Joseph D. Taylor.

**Funding acquisition:** Alain Nogaret.

**Investigation:** Joseph D. Taylor, Alain Nogaret.

**Methodology:** Alain Nogaret.

**Project administration:** Alain Nogaret.

**Software:** Joseph D. Taylor, Samuel Winnall.

**Supervision:** Alain Nogaret.

**Validation:** Samuel Winnall.

**Writing – original draft:** Alain Nogaret.

**Writing – review & editing:** Alain Nogaret.

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
