## [Decision Letter · Decision Letter 0]

4 May 2020

Dear Prof Nogaret,

Thank you very much for submitting your manuscript "Estimation of neuron parameters from imperfect observations" for consideration at PLOS Computational Biology.

As with all papers reviewed by the journal, your manuscript was reviewed by members of the editorial board and by several independent reviewers. In light of the reviews (below this email), we would like to invite the resubmission of a significantly-revised version that takes into account the reviewers' comments.

We cannot make any decision about publication until we have seen the revised manuscript and your response to the reviewers' comments. Your revised manuscript is also likely to be sent to reviewers for further evaluation.

Sincerely,

Hermann Cuntz

Associate Editor

PLOS Computational Biology

Daniele Marinazzo

Deputy Editor

PLOS Computational Biology

Reviewer's Responses to Questions

**Comments to the Authors:**

Reviewer #1: This fairly theoretical paper describes a process to estimate parameters of neuron models. The method is applied using surrogate data obtained from running a model of a rostral ventrolaterial medulla neuron. It contains several techniques to find back the global optimal estimate of the parameters of the model in question.

I think the paper is fairly well written, and rigorously describes the math behind the method.

However, in general I have two problems with the paper in its current form.

Firstly, although the method is described very rigorously, I find the results obtained rather less convincing. The method is only applied on a single model. For the results of that single model not many statistics are used. Quite some results are based on a single observation (as an example I'd give Fig 5). It would be good to show that the results are more robusted over more models and/or more trials.

Secondly, a big drawback of this paper is that it is based on surrogate data. Over the years many theoretical papers have been published with methods to optimize neuron parameters, but not many of these are actually used in real situations. The reason is that there is a world of difference between surrogate data and real experimental data. Just adding some noise to the surrogate data doesn't really represent the real world problems. Therefore I'd strongly suggest that the authors consider at least to add an example where they apply their method to real experimental data. I do understand that of course in that case some analysis can't be performed, because one doesn't know the original parameters, but it still allows for an analysis of number of solutions found, etc.

line 102: using a least-square error function has as big drawback that it is very sensitive to tiny shifts in timings. E.g. if an AP is shifted by a couple of ms, it's exact shape could still be the same, but the error could be huge. Could you discuss this in the paper, and how this could be solved.

line 193: at the moment you're adding noise to the voltage. I'd also suggest adding noise to the current, which would reflect reality better and which could lead to some shift of e.g. the AP timings etc.

line 279: it's not very clear to me how this percentage was obtained. As I mentioned above, I also think these numbers are too precise and for a particular case, I'd rather see mean/std of these over a couple of trials, use cases.

line 331: to say 'systematic' it has to be quantified better

line 352: typo: missing function 'of' a biological

line 415: I think, based on the fact that this is very theoretical work, it is far too much a stretch to start draw conclusion from this about brain function

line 515: about the adaptive time step approach in general. The NEURON simulator has an adaptive time step approach that does exactly this in a more advanced way (i.e. decrease the time step at times when more data points are necessary). It would make sense to reference and discuss that method.

About the choice of parameters. You do tune a lot of parameters at the same time. Especially the kinetic parameters can in generally be obtained from separate experiments / literature. It might make sense to discuss if it's better to fit everything at the same time, or break things up in different components.

Reviewer #2: Comments to authors attached as pdf.

Reviewer #3: Comments see attached document

**Have all data underlying the figures and results presented in the manuscript been provided?**

Reviewer #1: Yes

Reviewer #2: Yes

Reviewer #3: Yes

PLOS authors have the option to publish the peer review history of their article (what does this mean?). If published, this will include your full peer review and any attached files.

Reviewer #1: No

Reviewer #2: Yes: Robert P Gowers

Reviewer #3: No
---

## [Decision Letter · Decision Letter 1]

15 Jun 2020

Dear Prof Nogaret,

We are pleased to inform you that your manuscript 'Estimation of neuron parameters from imperfect observations' has been provisionally accepted for publication in PLOS Computational Biology.

Best regards,

Hermann Cuntz

Associate Editor

PLOS Computational Biology

Daniele Marinazzo

Deputy Editor

PLOS Computational Biology

Please make sure to address the final concerns of Reviewer #2.

Reviewer's Responses to Questions

**Comments to the Authors:**

Reviewer #1: The authors have addressed all the remarks.

Reviewer #2: Review uploaded as attachment.

Reviewer #3: Placeholder

**Have all data underlying the figures and results presented in the manuscript been provided?**

Reviewer #1: Yes

Reviewer #2: Yes

Reviewer #3: Yes

PLOS authors have the option to publish the peer review history of their article (what does this mean?). If published, this will include your full peer review and any attached files.

Reviewer #1: No

Reviewer #2: Yes: Robert Gowers

Reviewer #3: Yes: Alexander J Stasik

---

## [Editor Report · Acceptance letter]

9 Jul 2020

PCOMPBIOL-D-20-00043R1 

Estimation of neuron parameters from imperfect observations

Dear Dr Nogaret,

I am pleased to inform you that your manuscript has been formally accepted for publication in PLOS Computational Biology. Your manuscript is now with our production department and you will be notified of the publication date in due course.

With kind regards,

J&J Graphics
